
# Spatial separation of spin currents in transition metal dichalcogenides

Antonio L. R. Manesco[1*] and Artem Pulkin[1,2]

**1** Kavli Institute of Nanoscience, Delft University of Technology,
Delft 2600 GA, The Netherlands
**2** Qutech, Delft University of Technology, Delft 2600 GA, The Netherlands

⋆ am@antoniomanesco.org

## Abstract

We theoretically predict spatial separation of spin-polarized ballistic currents in transition metal dichalcogenides (TMDs) due to trigonal warping. We quantify the effect in terms of spin polarization of charge carrier currents in a prototypical 3-terminal ballistic device where spin-up and spin-down charge carriers are collected by different leads. We show that the magnitude of the current spin polarization depends strongly on the charge carrier energy and the direction with respect to crystallographic orientations in the device. We study the (negative) effect of lattice imperfections and disorder on the observed spin polarization. Our investigation provides an avenue towards observing spin discrimination in a defect-free time reversal-invariant material.

## 1 Introduction

Manipulating electronic degrees of freedom beyond electric charge, *e.g.*, spin, is a cornerstone for future electronic devices. The creation, control, and detection of spin-polarized currents

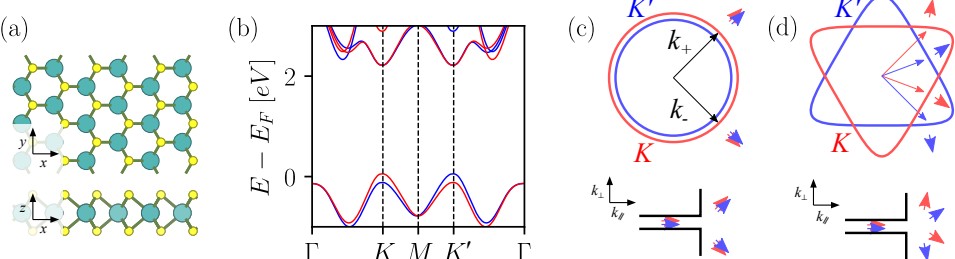

Figure 1: Atomic, electronic structure of monolayer $MoS_2$ and the concept of spin discrimination in transport based on band warping. (a) Crystal structure of $MoS_2$. Two hexagonal planes of S atoms sit at both sides of the middle hexagonal plane constituted by Mo atoms. A top view shows that the atoms are organized in a honeycomb crystal structure. (b) Spin-resolved $MoS_2$ band structure obtained from the tight-binding model. The spin-orbit coupling breaks spin degeneracy and causes spin-momentum locking as illustrated by colored (blue=spin-up; red=spin-down) bands. (c, d) Top: schematic illustration of 2D Fermi surfaces, charge carrier momenta (arrows) and their group velocities (short colored arrows) with (d) or without (c) trigonal band warping effect. Colors denote the coupled spin and valley of charge carriers. Bottom: a schematic illustration of charge carriers leaving a nanoconstraint towards a semi-infinite 2D plane. Depending on the trigonal warping effect, charge carriers with different spin and valley index are either aligned along the same set of directions (c) or become split (d). In the latter case, the geometry of a 2D device can be used to discriminate charge carriers with respect to their spin.

typically rely on combining different classes of materials in heterostructures, such as ferromagnets, semiconductors, and superconductors [1–3]. The need of combining different materials, however, imposes challenges such as the fabrication of high-quality interfaces [1–7]. Thus, it is desirable to find a single-material platform offering a full control of spin-polarized currents.

Past works predicted that it is possible to spatially separate charge carriers from different Fermi surface pockets in graphene [8–10]. At low-doping, the Fermi surface of graphene consist in two circles centered at the high-symmetry points $K$ and $K'$ [11]. Increasing charge density further leads to trigonal warping of the Fermi surface [11–13], as schematically shown in Fig. 1 (c, d). As a consequence, the quasiparticles' velocities at these two pockets (valleys) depend on lattice orientation. Thus, when injected from a quantum point contact, these quasiparticles split in two jets, one for each valley [8–10]. These coherent electron jets were recently observed in experiments performed with gate-defined bilayer graphene devices [14].

Unlike graphene, transistion metal dichalcogenides (TMDs) lack inversion symmetry, as shown in Fig. 1 (a), and present strong spin-orbit splitting effects in the electronic band structure [15–18]. The lack of inversion symmetry combined with strong spin-orbit coupling results in the gapped electronic band structure where spin degeneracy is lifted, shown in Fig. 1 (b) [15, 19]. Because of time-reversal symmetry, valence bands at high-symmetry points $K$ and $K'$ host electronic states with opposite spins [16]. Therefore, valley phenomena in TMDs must also have their spin counterpart [16–18].

In this work, we show that a combination of band structure properties in TMDs leads to the generation of transverse spin-polarized currents. While valley physics in monolayer graphene, supposedly, manifests itself only at large doping, in TMDs the effect is already pronounced within few hundreds meV inside valence band. We use a simplistic analytical picture together with scattering matrix transport simulations to estimate the spin polarization value as 50% (on a scale where 0% is spin-degenerate current and 100% is the maximal spin polarization value).

## 2  Spin polarization in monolayer MoS$_2$

We propose monolayer transition metal dichalcogenides (TMDs) such as MoS$_2$ shown in Fig. 1(a) as an alternative platform for observing warping-induced valley polarization. Because of the spin-valley coupling (Fig. 1(b)) the valley polarization of charge carriers causes spin imbalance. Unlike the valley index, spins can be detected unambiguously in optical and transport experiments.

The large spin-orbit coupling (SOC) together with the lack of particle-hole symmetry and broken spin degeneracy in the band structure of monolayer MoS$_2$ introduces sizeable corrections to the simplified picture of warping-induced valley and spin polarization. First, monolayer MoS$_2$ is a gapped material where spin-orbit induced level splitting and spin-valley coupling is much more pronounced in valence bands. Thus, for the rest of this section we consider the transport of hole charge carriers with energies $E$ counted from the valence band top. Second, the size of spin-orbit splitting is comparable to the energy scale of the trigonal warping effect. As a result, hole states within the valence band window for which spins are energetically split show not only valley, but also finite spin polarization, as illustrated in Figs. 2, and 3.

### 2.1  Effective model

We consider an effective model for TMDs to revisit the previous results for graphene systems [8–10]. Using the effective model from Ref. [20], we find an analytical expresion for the Fermi surface given by (the derivation is provided in [21])

$$k_\tau(\phi, E) = \sqrt{\frac{2E\Delta}{2\Delta\alpha - \gamma_3^2} + \tau\, \frac{2E\Delta\gamma_3\kappa}{\left(\gamma_3^2 - 2\Delta\alpha\right)^2}\cos(3\phi)}, \tag{1}$$

where $E$ is the relative energy from the top of the valence band, $\phi$ is polar angle of $\boldsymbol{k}_\tau$, $\tau$ is the spin-valley index, $\Delta = 1.66\,\text{eV}$ is the gap size, $\alpha = 1.72\,\text{eVÅ}$, $\gamma_3 = 3.82\,\text{eVÅ}$, $\alpha = 1.72\,\text{eVÅ}^2$, $\kappa = 1.02\,\text{eVÅ}^2$ [20].

Band warping makes charge carrier transport properties anisotropic. To demonstrate it, we consider a system where charge carriers are injected into a semi-infinite scattering region from a narrow lead with the width $W = 30a$ (where $a = 3.129\,\text{Å}$ is the lattice constant), as depicted in Fig. 1 (c,d). The charge carrier confinement across the lead results in the quantized component $k_\perp$ which, in turn, generates a finite number one-dimensional (1D) energy subbands. Each subband hosts 1D energy subband states in the form of standing waves across the lead. Possible standing wave lengths $\lambda_n$ are integer fractions of the total lead width $\lambda_n = L/n$, $n \in \mathbb{N}$. Since the momenta of each subband mode must lie in the Fermi surface, each mode has also a well-defined momentum $k_\parallel$ along the lead. Once injected into the scattering region, subband states couple to two-dimensional bulk band structure states at the same energy and matching quasi-momentum defined by $\boldsymbol{k}$ from Eq. 1. Zero or more pairs of transverse bulk modes may couple to a single standing wave with length $\lambda$. In the simplest case, the selection rule picks a pair of bulk-like Bloch states in the scattering region with quasi-momentae $\boldsymbol{k}_{\tau,n}^{(+)}$, $\boldsymbol{k}_{\tau,n}^{(-)}$ satisfying the following condition

$$\boldsymbol{k}_{\tau,n}^{(+)} - \boldsymbol{k}_{\tau,n}^{(-)} = \boldsymbol{e}_\perp \pi/\lambda_n, \tag{2}$$

where $\boldsymbol{e}_\perp$ is the unity vector perpendicular to the injection direction. For circular Fermi surface $k(\phi, E) = k_F(E)$, the above condition results in (mirror-)symmetric transport properties with respect to the lead direction for each spin/valley as schematically shown in Fig. 1 (c). Consecutively, the loss of mirror symmetry within the Fermi surface of each individual valley (illustrated in Fig. 1(d)) will open a possibility for valley- and spin-selective transport.

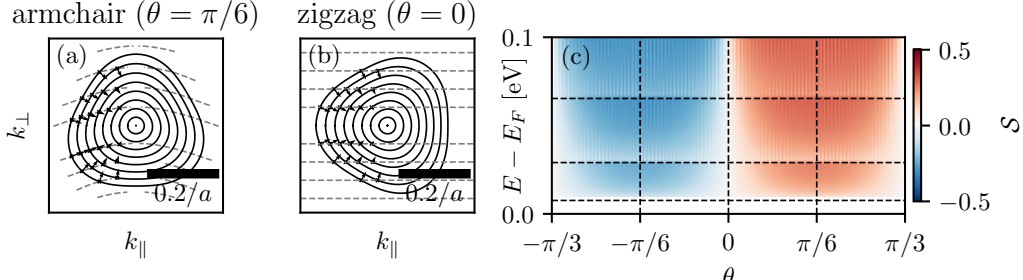

Figure 2: Spin polarization of ballistic currents induced by warping of the Fermi surface. (a, b) Illustration of wave vector selection rules for armchair (a) and zigzag (b) nanoconstraints. The closed lines correspond to Fermi surfaces ("electron pockets") at different energies $E - E_F$ in increasing order from $E = E_F$ to $E - E_F = 0.1\,\text{meV}$. The arrows show the direction of the velocity for each of the injected states at selected $k_\parallel$ of the lead mode. (c) Computed spin polarization $\mathcal{S}$ for different crystallographic orientations $\theta$ of the nanoconstraint. The angle $\theta$ is measured with respect to zigzag direction. The vertical lines indicate armchair and zigzag orientations, whereas the horizontal lines indicate the population of extra subbands.

To test the above concept of warping-induced spin polarization, we compute the polarization resulting from the low-energy model with Fermi surface given by Eq. 1. According to Eq. 1, trigonal warping effect is more pronounced for charge carriers further away from $E_F$. Thus, we expect a strong energy dependence of the spin polarization $\mathcal{S}$. Since the charge carrier current value is proportional to the group velocity of charge carriers, we define $\mathcal{S}$ as

$$\mathcal{S} = \frac{\sum_{n,\tau} \tau \left( \boldsymbol{v}_{n,\tau}^{(+)} + \boldsymbol{v}_{n,\tau}^{(-)} \right)}{\sum_{n,\tau} \left( \boldsymbol{v}_{n,\tau}^{(+)} + \boldsymbol{v}_{n,\tau}^{(-)} \right)}, \quad \boldsymbol{v}_{n,\tau}^{(\pm)} = \partial_{\boldsymbol{k}} E_\tau(\boldsymbol{k}_{n,\tau}^{(\pm)}). \tag{3}$$

The spin polarization $\mathcal{S}$ plotted in 2(c) indeed confirms this behavior: asymptotically zero polarization at small energies increases towards larger values deeper into valence bands. From Figs. 2 (a) and (c), it is also evident that shifting the Fermi level towards additional subbands results in an abrupt drop in the total spin polarization. The polarization is recovered as the doping increases even more, saturating at approximately $\sim 50\%$.

One can intuitively understand the saturation of spin polarization by imagining the case when the Fermi surface is almost a perfect triangle. If we consider that the injection lead is much larger than $\lambda_F$, the Fermi surface region with $v_\parallel > 0$ is fully populated. Thus, as one can observe from Fig. 2 (a), the spin/valley polarization depends on the ratio between the Fermi surface area corresponding to quasiparticles moving with $v_\perp > 0$ and $v_\perp < 0$. For a nearly perfect triangular Fermi surface, this ratio is $\approx 0.5$. It is however worth mentioning that 50% of polarization is saturation value for this particular geometry. We do not explore whether geometries more involved than the ones considered in this manuscript could lead to higher polarization values.

Polarization values are also affected by the crystallographic orientation of the lead. If the nanoconstraint is parallel to the "zigzag" direction, both hole pockets are mirror symmetric with respect to the $k_\perp$ direction (see Fig. 2 (b)). The spin polarization is zero in this case, also shown in Fig. 2 (c). We show, however, that this case is a sweetspot: for all other orientations we expect no mirror symmetry of the Fermi surface, thus, a finite spin polarization. In Fig. 2 (c) we show the spin polarization as a continuous function of the angle $\theta$ with $\theta = 0$ being the zigzag orientation. The computed polarization values are anti-symmetric with respect to this case.

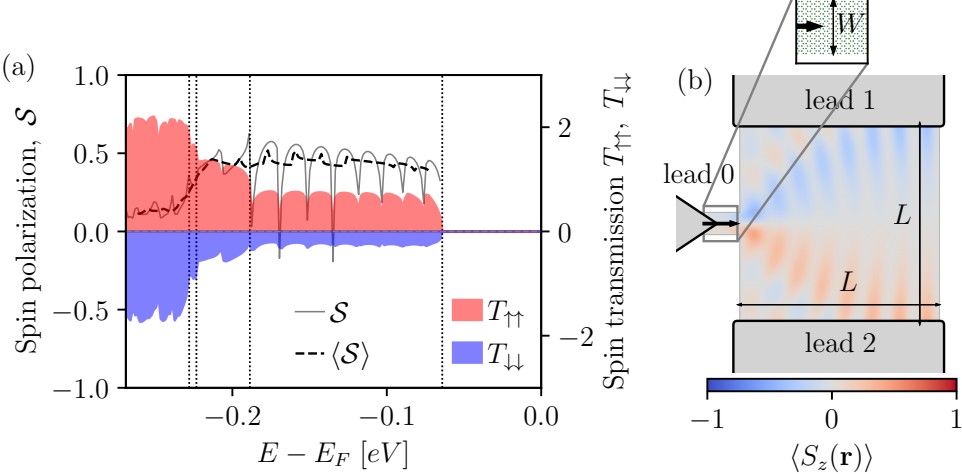

Figure 3: (a) Two-terminal spin polarization $\mathcal{S}$ of the device shown in panel (b). The polarization is computed for states transmitted from lead 0 to lead 1 (black line) and rolling average $\langle\mathcal{S}\rangle$ taken every 10 points (dashed line) show a saturation as a function of charge carrier energy $E$ close to 50%. Lines plot spin polarization (with values on the left axis) while individual contributions $T_{\uparrow\uparrow}$, $T_{\downarrow\downarrow}$ are shown in color and correspond to the axis on the right edge. The vertical dotted lines indicate the top of the subbands at $k_\parallel = 0$. (b) Scheme of the 3-terminal device used for the tight-binding transport simulations. The scattering region spans a square area of size $L \times L$, $L = 150a$. We also show the spin density $\langle S_z(\mathbf{r})\rangle$ for $E = 50\,\mathrm{meV}$, for which spin-polarized hole jets are visible.

The described valley-dependent phenomenon was previously observed in graphene devices in the form of ballistic charge carrier current jets [8–10, 14]. While the observation of current jets does not necessarily impose a regime with Fermi surface warping and valley polarization, it does manifest the formation of subbands and the selection rule in Eq. 2, thus becoming a strong argument in favor of the possibility of valley-polarized currents in the setup. We emphasize, that the spin/valley polarization effect depends strongly on both the nanoconstraint orientation and the Fermi level. At the same time, in the limit of a wide nanoconstraint, the spin/valley polarization remains finite and does not depend on edge physics and size quantization.

## 2.2 Tight-binding simulations of multiterminal devices

To quantify band structure effects on the spin polarization in monolayer $MoS_2$, we performed tight-binding simulations of the 11-band tight-binding model of $MoS_2$ [19] using Kwant [22] software package. We simulated a 3-terminal T-shaped device depicted in Fig. 3 (a) to quantify these effects. In this device, electrons are injected from an armchair quantum point contact lead with width $W = 10a$ into a scattering region of finite but large enough size $L \times L$, $L = 150a$, to ensure our results are not governed by edge physics. We compute the spin polarization between leads $i$ and $j$ as:

$$\mathcal{S} = \frac{T_{\uparrow\uparrow}^{ij} - T_{\downarrow\downarrow}^{ij}}{T_{\uparrow\uparrow}^{ij} + T_{\downarrow\downarrow}^{ij}},\qquad(4)$$

where $T_{\sigma\sigma'}^{ij}$ are the transmission coefficients from lead $i$ from lead $j$ for scattering states entering the scattering region with spin $\sigma$ and leaving the scattering region with spin $\sigma'$. The Eq. 4

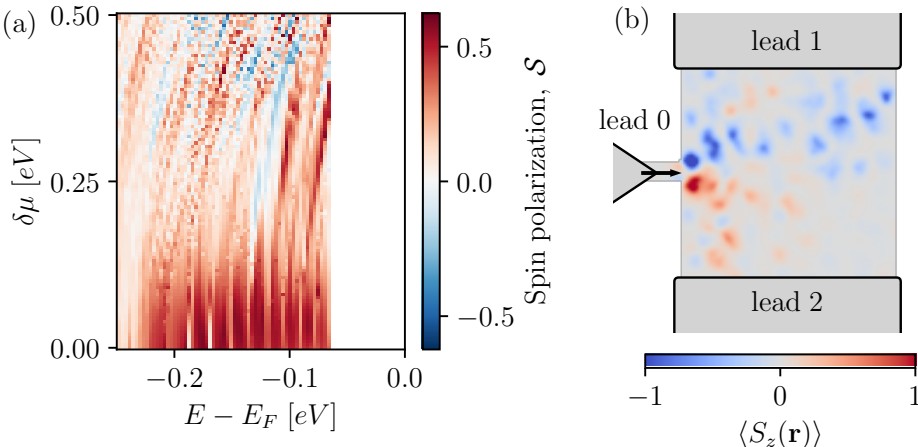

Figure 4: (a) Effects of disorder on the spin transmission in the tight-binding model of monolayer $MoS_2$. The polarization is rapidly suppressed for $\delta\mu \approx 0.1\,\text{eV}$. (b) Spatial map of $\langle S_z \rangle$ showing the breakdown of the spin jets for $\delta\mu = 0.1\,\text{eV}$ and $E - E_F = -0.1\,\text{eV}$.

contains only spin-conserving transmission coefficients with $\sigma = \sigma'$ because the spin along the $z$-direction is conserved.

Fig. 3(b) demonstrates spin-resolved transmission and net spin polarization as a function of charge carrier energy $E$. The spin polarization quickly grows and saturates at $\mathcal{S} \approx 0.5$ before leaving the SOC energy window. We attribute this observation to significantly less dispersive energy bands of $MoS_2$ accompanied by the large band gap value. Because of this, the trigonal warping is developed in a narrow energy window and easy to reach even when the spin-orbit energy splitting constraint is in place. The occupation of additional subbands leads to steps in transmission and polarization values as described previously. The spatially-resolved local current polarization shown in Fig. 3(a) demonstrates spin-polarized hole jets, similar to the ones predicted and observed in graphene [8–10, 14].

Since valley conservation requires the absence of large momentum scattering, valley polarization is fragile against short-range disorder [23, 24]. If intravalley scattering is enabled by disorder, the Fermi surface is randomly occupied, and the spin polarization is suppressed. We demonstrate this behaviour by adding a random uncorrelated onsite potential uniformly distributed as $\delta\mu_i \in [-\delta\mu, \delta\mu]$ at all atomic sites $\mathbf{r}_i$ inside the scattering region. The suppression of spin polarization is demonstrated in Fig. 4(a). As the disorder strength increases the average spin polarization decreases and the sign fluctuates as a function of the energy $E$. The local spin density map shown in Fig. 4(b) illustrates the effects of disorder: the spin-polarizaed hole jets shown in Fig. 3(a) vanish in the presence of disorder. Thus, the regime described here should hold as long as the system's dimensions are much smaller than the intra- and intervalley scattering lengths. In fact, the spin polarization requires ballistic samples regardless of the device geometry.

We consistently observe spatial separation of spin currents for other device geometries. This separation is a direct result of the asymmetry between the two Fermi pockets. To support this claim, we consider a cross-shaped 4-terminal device with two armchair and two zigzag leads, shown in Fig. 5. All leads in this devices are wide enough ($W = 50a$) such that many propagating modes are present and thus the hole jets shown in Fig. 3(a) are not evident.

We first consider the case in which current is injected through an armchair lead due to the similarity to the 3-terminal device presented. The two leads adjacent to the source lead

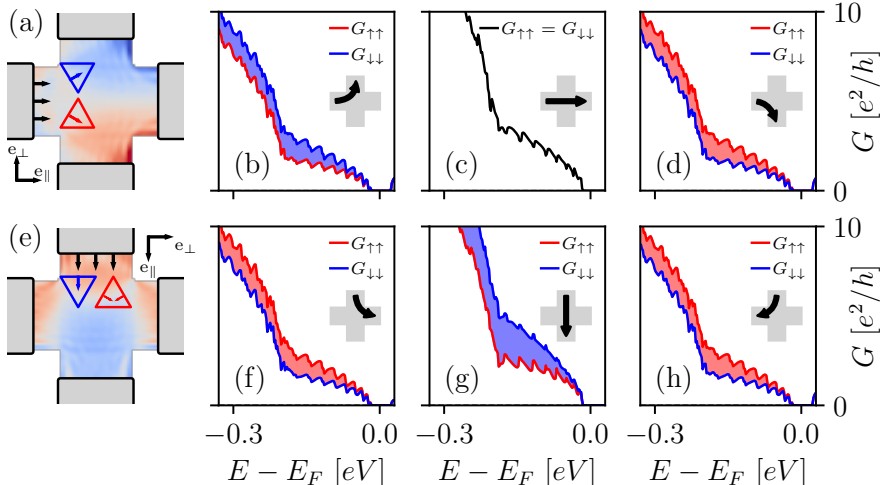

Figure 5: Spin polarization and spin density maps for the same 4-terminal device where the source lead is parallel to the zigzag (a) or the armchair (b) direction. Current is injected from the leads indicated by the black arrows. Individual panels (b-d) and (f-h) show the spin-resolved transmission of current leaving each respective device lead. Current is injected along the $\mathbf{e}_\parallel$ direction, and the transverse direction is denoted by $\mathbf{e}_\perp$. The Fermi pockets are illustrated as colored triangles: red triangles indicate spin-up Fermi pocket, whereas blue triangles indicate spin-down. Fermi velocity the injected leads are shown as arrows perpendicular to the Fermi pockets.

measure spin-polarized current, with opposite spin polarizartions [Fig. 5 (b) and (d)]. The spin polarization saturates at ∼ 25%. The current measured at the other armchair lead at the opposite side of the device is not spin-polarized [Fig. 5 (c)]. This behavior is consistent with the orientation of trigonally warped Fermi surfaces which tend to diverge the injected spin currents sideways.

When the injection lead is oriented along the zigzag direction in a 4-terminal device, the spin polarization does not resemble the 3-terminal case. The reason is visible from the Fermi pockets and Fermi velocities shown in Fig. 5 (e). The Fermi velocity direction shows a spin-up net current along both $\pm\mathbf{e}_\perp$ [Figs. (f) and (h)], while for spin-down the Fermi velocity is collimated along the $\mathbf{e}_\parallel$ direction [Figs. (g)]. As a reasult, spin-up current (blue) tends to propagate across the device towards the opposite lead while spin-down current (red) splits and diverges towards both side leads.

## 3 Summary

We developed a theory of valley transport for trigonally-warped band structures found in graphene and 2D transition metal dichalcogenides. Our analysis showed that transverse valley-polarized electron currents are expected to exist for all crystal orientations, except when the current is injected perfectly along the zigzag direction. In TMDs, spin-valley locking together with a large gap due to inversion symmetry breaking enable the formation of spin-polarized current jets at low-doping with holes. If current is injected through a thin nanoconstraint, we observe the formation of spin-polarized hole jets, analogous to the valley-polarized electron jets recently observed in graphene. Both our effective model and, more realistic, 11 orbital tight-binding calculations show a spin/valley polarization as high as ∼ 50% in 3-terminal

devices. Furthermore, we show that the phenomena is also observed in 4-terminal devices. We evaluated the role of the disorder and estimated the critical disorder strength for the device considered.

## Acknowledgements

The authors thank Anton Akhmerov and Daniel Varjas for useful discussions.

**Author contributions**  A.P. initiated the project, and developed the analytical model. A.L.R.M. performed the MoS$_2$ tight-binding calculations, and identified the trigonal warping effects. The authors designed the devices simulated and wrote the manuscript jointly.

**Funding information**    This work was supported by VIDI grant 016.Vidi.189.180.

## Data availability

The code and data used to produce the figures and derive the effective model are available in Ref. [21].

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
