# Peer review of "Spatial separation of spin currents in transition metal dichalcogenides"

_SciPost Physics, doi:SciPost Phys. Core 6, 036 (2023)_

## Round 2 · Referee Report · Anonymous (Referee 1) · 2022-11-7

Strengths

1. The mechanism of current-induced valley polarization is discussed with an apparent and minimum symmetry argument.
2. The result is applied to TMD and discovers a spin polarization associated with valley polarization.
3. Effect of disorders is discussed and may arouse experimental effort.

Weaknesses

The importance of trigonal warping of the Fermi surface is not fully addressed, e.g., angle dependence in the proposed device.

Report

Antonio et al. report a theoretical prediction and numerical simulation that the current injection of a ballistic TMD or Graphene device will lead to a spatial separation of carrier spin due to trigonal warping. An optical counterpart of such a current-induced spin polarization is proposed in Phys. Rev. Lett. 108, 196802
(cited) and further explored in PNAS 117 (29) 16749-16755 (uncited) with time-reversal symmetry(TRS) broken cases, which shows that in the thermodynamic limit, a simple symmetric argument forbids current-induced spin polarization. However, when the system couples with a confined lead and finite size effect happen due to trigonal warping, such a phenomenon is allowed, proposed by the authors. The authors adopted real-space simulation and addressed how the disorder will affect such spatial splitting of spin current. I believe this work would attract broad interest in the mesoscopic transport and device physics area and deserves publication with some minor changes requested below.

Requested changes

1. In figure3 b, in the point contact region(10a region), what's the orientation of the lattice? Or is an effective model on square lattice used? Providing these details can help to understand the angle dependence of the jet.
2. Does the atomic termination of lead0 play a role?
3. How does the disorder affect 4 terminal transport?
4. case (e) and (g) in Figure.5 is quite interesting. Will the long arm in the e// direction enhance the ratio of spin-down polarization?

  • validity: top
  • significance: high
  • originality: high
  • clarity: top
  • formatting: excellent
  • grammar: perfect

Author:  Antonio Manesco  on 2023-02-17  [id 3366]

(in reply to Report 1 on 2022-11-07)

Antonio et al. report a theoretical prediction and numerical simulation that the current injection of a ballistic TMD or Graphene device will lead to a spatial separation of carrier spin due to trigonal warping. An optical counterpart of such a current-induced spin polarization is proposed in Phys. Rev. Lett. 108, 196802 (cited) and further explored in PNAS 117 (29) 16749-16755 (uncited) with time-reversal symmetry(TRS) broken cases, which shows that in the thermodynamic limit, a simple symmetric argument forbids current-induced spin polarization. However, when the system couples with a confined lead and finite size effect happen due to trigonal warping, such a phenomenon is allowed, proposed by the authors. The authors adopted real-space simulation and addressed how the disorder will affect such spatial splitting of spin current. I believe this work would attract broad interest in the mesoscopic transport and device physics area and deserves publication with some minor changes requested below.

We thank the referee for the feedback. We also thank the referee for pointing out a relevant reference that we originally missed.

We considered all the minor changes suggested by the referee and address them in the manuscript. We answer all the suggestions below:

1. In figure3 b, in the point contact region(10a region), what's the orientation of the lattice? Or is an effective model on square lattice used? Providing these details can help to understand the angle dependence of the jet.

Indeed, Figure 3 was computed using the tight-binding model from reference [18], as we mentioned in the third paragraph of Sec. 3. We missed mentioning the lattice orientation used for this simulation, and now we properly mention that the injection direction is armchair. We also updated Fig. 3 accordingly.

2. Does the atomic termination of lead0 play a role?

No, the atomic terminations of lead0 play no role in our calculations. There are no edge states within the energy window that we explored.

3. How does the disorder affect 4 terminal transport?

Disorder affects the spin polarization irrespective of the device geometry. The reasoning behind it is that the effects that we reported rely on the Fermi velocity of each injected state. With disorder, elastic intravalley scattering occurs and, as a consequence, the velocities are randomized over the Fermi surface. As a consequence, spin polarization vanishes. This explanation was provided in the fifth paragraph of Sec. 3 and is independent of device geometry. We also emphasize now that the effects of disorder are independent on the device geometry.

4. case (e) and (g) in Figure.5 is quite interesting. Will the long arm in the e// direction enhance the ratio of spin-down polarization?

We do not expect a longer arm to enhance the ratio for two reasons: (i) the polarization values in Fig. 6 are very similar to the ones with a quantum point contact (Fig. 3); (ii) the transverse velocity develops right after the electrons leave the arm (one can perhaps see this in Figs. 5(a, e)). Perhaps an increase of the region in the middle might enhance the maximum polarization, but for the simulations considered, since we already reached a value close to the theoretical maximum (see the answer to the second item of the Anonymous Report 3), we do not expect significant effects by changing the device geometry.

---

## Round 2 · Referee Report · Anonymous (Referee 2) · 2022-11-21

Report

I have reviewed the paper entitled “Spatial separation of spin currents in transition metal dichalcogenides ”, by Antonio L. R. Manesco, et. all. The paper investigates the effect of trigonal warping in the creation of spin-polarized currents in multi-terminal device-based monolayer MoS2.

My overall assessment of the paper is positive, the presentation is sound clear, but some relevent previous literature is not cited. Above all, to further credit the author, I believe the presented main result - i.e. the discovery of spin-polarized current - is correct.

In principle, the problem is interesting and relevant to the readership of Scipost, but I believe that the theory presented is based on graphene, which is already discussed in ref.[8], could be replaced with the low-energy effective model of MoS2, for instance, the model introduced in paper “J. Phys.: Condens. Matter 28 (2016) 495001”. With this, the manuscript will fit to be published on SciPost Physics. Otherwise, I recommend publication of this work in SciPost Phys. Core.

  • validity: good
  • significance: good
  • originality: good
  • clarity: high
  • formatting: good
  • grammar: excellent

Author:  Antonio Manesco  on 2023-02-17  [id 3367]

(in reply to Report 2 on 2022-11-21)

I have reviewed the paper entitled “Spatial separation of spin currents in transition metal dichalcogenides ”, by Antonio L. R. Manesco, et. all. The paper investigates the effect of trigonal warping in the creation of spin-polarized currents in multi-terminal device-based monolayer MoS2.

My overall assessment of the paper is positive, the presentation is sound clear, but some relevent previous literature is not cited. Above all, to further credit the author, I believe the presented main result - i.e. the discovery of spin-polarized current - is correct.

In principle, the problem is interesting and relevant to the readership of Scipost, but I believe that the theory presented is based on graphene, which is already discussed in ref.[8], could be replaced with the low-energy effective model of MoS2, for instance, the model introduced in paper “J. Phys.: Condens. Matter 28 (2016) 495001”. With this, the manuscript will fit to be published on SciPost Physics. Otherwise, I recommend publication of this work in SciPost Phys. Core.

We thank the referee for the valuable feedback. We also apologize for missing any relevant literature. We added a few more references that we found useful during the review of this submission. We hope the new additions were sufficient. And if the referee has specific references that should be consider, we kindly ask they to please mention them.

We agree with the referee that in Sec. 2 we essentially reviewed what was done for graphene. We initially considered it because we realized that some aspects were not explicitly stated in the previous works, and we had positive feedback from experimentalists in the field for clarifying those. Particularly, we could not find a clear description of the selection rule in Eq. 3 in the previous works.

However, after considering the report, we realized that starting with a model for MoS$_2$ could still be presented in a simplified manner. And therefore we now start from scratch a discussion on MoS$_2$. We believe that this change will not influence on the understanding of the effective model and also directly address the material platform we consider.

We hope that with this addition the editor-in-charge will consider the referee recommendation to publish on SciPost Physics.

---

## Round 2 · Referee Report · Anonymous (Referee 3) · 2022-12-3

Strengths

1- Hot topic of 2D materials at the border between fundamental properties and potential applications (spintronics)

2- The Tight-Binding model for MoS2 take account the different types of relevant orbitals in this material.

Weaknesses

1- The possible effects of edge states are not discussed.

2- The presented calculations show a spin/valley polarization as high as ∼ 50% in 3-terminal devices. But the it is not clear if this value is related to the geometry of the device, or if it is theoretically a maximum possible value?

Report

The proposed manuscript presents a numerical study of the spin current in Transition Metal Dichalcogenides (TMDC). It shows how it is possible to combine the effects of the spin-orbit coupling (strong in TMDC) and trigonal band warping effect to spatially separate spin-polarized ballistic currents in a nanodevice. The subject seems to me very current and relevant in the context of 2D materials and their applications to spintronics. The numerical methods used are relevant and the results of the calculations are clearly presented. The authors propose two different approaches. First, an effective model to describe the electronic bands of graphene, and second, a more realistic approach based on a Slater-Koster Tight-Binding Hamiltonian to describe the case of MoS2 (TMDC). The first one is interesting, but in my opinion the second one is much more relevant, since it takes into account the nature of the different orbitals of metal atoms and chalcogen atoms. These two approaches give similar results. The manuscript is well presented, it concerns a hot topic of 2D materials at the border between fundamental properties and potential applications, so I think that should be published in SciPost Physics Core.

Requested changes

I have no major change request, only 3 questions/suggestions:

(1) In Sec. 3, the authors use a fairly large system to avoid edge effects. However, in these type of 2D materials, transport through edge states can be important. Is it possible to comment on this, or to show that increasing the size does not change significantly the result.

(2) The Tight-Binding calculations show a spin/valley polarization as high as ∼ 50% in 3-terminal devices. I wonder if this "50%" value is related to the geometry of the device under study or is it close to a theoretical maximum value that could not be exceeded in any device?

(3) The Slater-Koster Tight-Binding used to simulate MoS2 is well known, however for the clarity of the paper, it should be described a bit more. Besides, I do not understand why the authors call it “the 8-band tight-binding model” while there are in fact 11 orbitals per cell.

  • validity: high
  • significance: top
  • originality: top
  • clarity: high
  • formatting: good
  • grammar: -

Author:  Antonio Manesco  on 2023-02-17  [id 3368]

(in reply to Report 3 on 2022-12-03)

The proposed manuscript presents a numerical study of the spin current in Transition Metal Dichalcogenides (TMDC). It shows how it is possible to combine the effects of the spin-orbit coupling (strong in TMDC) and trigonal band warping effect to spatially separate spin-polarized ballistic currents in a nanodevice. The subject seems to me very current and relevant in the context of 2D materials and their applications to spintronics. The numerical methods used are relevant and the results of the calculations are clearly presented. The authors propose two different approaches. First, an effective model to describe the electronic bands of graphene, and second, a more realistic approach based on a Slater-Koster Tight-Binding Hamiltonian to describe the case of MoS2 (TMDC). The first one is interesting, but in my opinion the second one is much more relevant, since it takes into account the nature of the different orbitals of metal atoms and chalcogen atoms. These two approaches give similar results. The manuscript is well presented, it concerns a hot topic of 2D materials at the border between fundamental properties and potential applications, so I think that should be published in SciPost Physics Core.

We thank the referee for the feedback and for the recommendation to publish the manuscript.

I have no major change request, only 3 questions/suggestions:

1. In Sec. 3, the authors use a fairly large system to avoid edge effects. However, in these type of 2D materials, transport through edge states can be important. Is it possible to comment on this, or to show that increasing the size does not change significantly the result.

Indeed, edge states could play a significant role. However, as we mention in the reply to the contributed report, there are no edge states within the energy window considered in this work.

Regarding the size-dependent simulations: we already performed the calculation which thin (Fig. 3) and wide (Fig. 5) leads and see no significant difference. And, in fact, the polarization values are compatible -- in both, we achieve a polarization of $\sim 50\%$.

2. The Tight-Binding calculations show a spin/valley polarization as high as ∼ 50% in 3-terminal devices. I wonder if this "50%" value is related to the geometry of the device under study or is it close to a theoretical maximum value that could not be exceeded in any device?

For this particular geometry, $50\%$ of valley/spin polarization is indeed the theoretical maximum. To understand why we refer to Fig. 2(a) in the manuscript. As shown in Fig. 2(c), this is the direction with maximal spin polarization. The left side of the Fermi surface contains the hole states with positive velocity along $k_{\parallel}$. And one can also observe that there are ~2 times more states in the region with positive velocity along $k_{\perp}$ than the negative velocity region. This sets $50\%$ as an upper bound for spin polarization.

We emphasize that the reasoning above is only valid for the 3-terminal geometry. It is unclear to the authors whether more involved geometries could lead to higher polarization.

We thank the referee for this question and properly mention in the manuscript a comment on the theoretical maximal spin polarization.

3. The Slater-Koster Tight-Binding used to simulate MoS2 is well known, however for the clarity of the paper, it should be described a bit more. Besides, I do not understand why the authors call it “the 8-band tight-binding model” while there are in fact 11 orbitals per cell.

We add a few more details to the description of the Slater-Koster tight-binding model. We do not provide the complete Hamiltonian because its many parameters are explicitly written in Ref. [18], and constructed numerically in our code on Zenodo [22].

The referee is right that there are 11 orbitals per cell. We followed the description in reference [18] where the number 8 comes from 5 $d$-orbitals from Mo and 3 $p$-orbitals from each S atom. We agree that we should call it an 11-orbital model and fix the manuscript accordingly.

---

## Round 3 · Referee Report · Anonymous (Referee 4) · 2023-2-18

Report

The authors made some efforts to include my previous comment in favour of publishing in SciPost Physics instead of SciPost Physics Core. I do appreciate their effort for the appropriate effective model to describe their numerical predictions. Although their calculation is correct, after checking carefully the SciPost Physics Acceptance criteria, I am afraid that in spite of these extensions, this is still not sufficient for SciPost Physics, but rather remains at the level of SciPost Physics Core. . Accordingly, I recommend publication in Core

---

## Round 3 · Referee Report · Anonymous (Referee 5) · 2023-3-8

Report

The authors answered well to the small questions I had asked them. The new manuscript is well presented, coherent, and deserves to be published. These theoretical results are stimulating and should lead to experimental realizations, but they are still quite speculative and the experimental feasibility is still to be demonstrated, that is why it seems to me that this paper is more appropriate for SciPost Physics Core than for SciPost Physics.

---

## Round 3 · Author Response

Dear Editor,

We considered all the referee reports with their minor change requests. We believe that we now address all their suggestions, and therefore we expect that the new manuscript is suitable for publication.

One of the invited referees suggested focusing on the phenomenon in transition metal dichalcogenides, instead of revisiting the theory for graphene. We made the appropriate adjustments in the manuscript to now consider a continuum model for MoS2 in section 2. With this suggestion, the referee argued that our manuscript is suitable for publication in SciPost Physics, instead of SciPost Physics Core. We ask the editor and the referees to consider this change.

We provide extended answers to the referee reports.

---

## Round 3 · List of Changes

* Changed the continuum model from graphene to MoS2.
* Update figures 2 and 3.
* Implemented other minor changes as suggested by the referees.

---

## Editorial Decision

published